# Examination of Prognostic Factors Affecting Long-Term Survival of Patients with Stage 3/4 Gallbladder Cancer without Distant Metastasis

**DOI:** 10.3390/cancers12082073

**Published:** 2020-07-27

**Authors:** Ryota Higuchi, Takehisa Yazawa, Shuichirou Uemura, Yutaro Matsunaga, Takehiro Ota, Tatsuo Araida, Toru Furukawa, Masakazu Yamamoto

**Affiliations:** 1Department of Surgery, Institute of Gastroenterology, Tokyo Women’s Medical University, 8-1 Kawada-cho, Shinjuku-ku, Tokyo 162-8666, Japan; higuchi.ryota@twmu.ac.jp (R.H.); yazawa.takehisa@twmu.ac.jp (T.Y.); uemura.shuichiro@twmu.ac.jp (S.U.); matsunaga.yutaro@twmu.ac.jp (Y.M.); 2Department of Surgery, Ebara Hospital, 4-5-10 Higashiyukigaya, Ota-ku, Tokyo 145-0065, Japan; otakehiro72@yahoo.co.jp; 3Department of Surgery, Division of Gastroenterological Surgery, Tokyo Women’s Medical University, Yachiyo Medical Center, 477-96 Shinden, Oowada, Yachiyo-shi, Chiba 276-8524, Japan; araida.tatsuo@twmu.ac.jp; 4Department of Investigative Pathology, Tohoku University Graduate School of Medicine, 2-1 Seiryomachi, Aoba-ku, Sendai 980-8575, Japan; toru.furukawa@med.tohoku.ac.jp

**Keywords:** gallbladder cancer, distant metastases, prognostic factor, overall survival, surgical outcome

## Abstract

In advanced gallbladder cancer (GBC) radical resection, if multiple prognostic factors are present, the outcome may be poor; however, the details remain unclear. To investigate the poor prognostic factors affecting long-term surgical outcome, we examined 157 cases of resected stage 3/4 GBC without distant metastasis between 1985 and 2017. Poor prognostic factors for overall survival and treatment outcomes of a number of predictable preoperative poor prognostic factors were evaluated. The surgical mortality was 4.5%. In multivariate analysis, blood loss, poor histology, liver invasion, and ≥4 regional lymph node metastases (LNMs) were independent prognostic factors for poor surgical outcomes; invasion of the left margin or the entire area of the hepatoduodenal ligament and a Clavien–Dindo classification ≥3 were marginal factors. The analysis identified outcomes of patients with factors that could be predicted preoperatively, such as liver invasion ≥5 mm, invasion of the left margin or the entire area of the hepatoduodenal ligament, and ≥4 regional LNMs. Thus, the five-year overall survival was 54% for zero factors, 34% for one factor, and 4% for two factors (*p* < 0.05). A poor surgical outcome was likely when two or more factors were predicted preoperatively; therefore, new treatment strategies are required for such patients.

## 1. Introduction

Gallbladder cancer (GBC) is a rare disease [1]; it has fewer chemotherapy regimens than other cancers, and surgery is the only curative treatment that can be expected to confer long-term survival in patients [2]. Radical resection is also recommended for patients with advanced GBC, such as stage 3 or 4. In the 8th American Joint Committee Classification (AJCC) for GBC [3], tumors that perforate the serosa, directly invade the liver, or invade an adjacent organ structure such as the stomach, duodenum, colon, pancreas omentum, and extrahepatic bile ducts are classified as T3 stage, whereas tumors that invade the main portal vein, hepatic artery, or two or more extrahepatic organs or structures are classified as T4 stage. Due to the wide variety of developmental processes, including liver infiltration, hepatoduodenal infiltration, and lymph node metastasis [3,4], multiple modes of hepatectomy with lymph node dissection, bile duct resection, and sometimes a pancreaticoduodenectomy (PD) are required for radical resection. However, determinants such as the tumor progression factor frequency that defines T3 or T4 staging, identification of strong prognostic factors, and surgical outcome in patients with multiple prognostic factors remain to be investigated. In patients with stage 3/4 GBC without distant metastases, even if radical resection is obtained, the surgical outcome may be poor if multiple prognostic factors are observed.

Therefore, in this study, we have examined the frequency of local progression factors that define the T3 or T4 staging, which include strong prognostic factors, as well as the surgical outcomes of patients with multiple prognostic factors.

## 2. Results

### 2.1. Frequency and R0 Resection Rate Based on Local Factors of the Primary Tumor Site in Patients with Stage 3/4 GBC without Distant Metastasis

The incidence rates of local factors of the primary tumor site are shown in Table 1. The R0 resection rate was found to be low in patients with high hepatoduodenal ligament (HDL) invasion, pancreatic invasion, invasion in two or more extrahepatic organs or structures, and major vascular invasion.

### 2.2. Long-Term Outcomes Based on the Degree of Liver and HDL Invasion

The patients showed five-year survival rates of 48% when no invasion was present, 40% for direct invasion to the liver parenchyma <5 mm, 23% for direct invasion to the liver parenchyma ≥5 mm and <20 mm, and 14% for direct invasion to the liver parenchyma ≥20 mm (Appendix A, Appendix A). Because five-year survival rates of direct liver invasion of ≥5 mm and <20 mm, and ≥20 mm were similarly poor, they were used in subsequent analyses as invasion of 5 mm or more in the same group.

Furthermore, the five-year survival rate was 49% for no invasion, 41% for invasion to the right margin of the HDL, 16% for invasion to the left margin of the HDL, and 19% for invasion through the HDL (Appendix A and Appendix A). Because five-year survival rates of the invasion to the left margin of the HDL and through the HDL were similarly poor, they were used in subsequent analyses as the invasion of the left margin or entire HDL in the same group.

### 2.3. Prognostic Factors in Patients with Stage 3/4 GBC without Distant Metastasis

As summarized in Table 2, the univariate and multivariate analyses of the risk factors for overall survival showed that blood loss ≥1400 g (vs. <1400 g, hazard ratio [5]: 2.19), histologically poorly differentiated tumors or others (vs. well- or moderately-differentiated, HR: 2.26), liver invasion ≥5 mm (vs. no invasion, HR: 2.42), ≥4 regional lymph node metastases (vs. no lymph node metastasis [6], HR: 2.25), and treatment without adjuvant chemotherapy (vs. with, HR: 1.93) were independent risk factors. Invasion of the left margin or the entire area of the HDL (vs. no invasion, HR: 1.64, *p* = 0.053) and postoperative morbidity Clavien–Dindo Classification ≥3 (vs. ≤2, HR: 1.64, *p* = 0.054) were marginally insignificant.

### 2.4. Overall Survival Rate Based on the Number of Prognostic Factors Predicted Preoperatively

Liver invasion of 5 mm or more, left marginal invasion or invasion of the entire HDL, and metastases of four or more regional lymph nodes were considered predictable prognostic factors, to some extent, before surgery (Figure 1 and Appendix A). The five-year survival rate was 54.2% for zero factors, 34.2% for one factor, 5.87% for two factors, and 0% for three factors.

### 2.5. Characteristics and Recurrence in Groups Based on the Number of Prognostic Factors

Surgeries with a greater burden, such as hepatectomies of ≥two sections, bile duct resection or PD and vascular resection, were selected for radical resection in patients with a large number of prognostic factors predicted preoperatively. The analysis indicated that the R0 resection rate was lower for two or three prognostic factors than that for the other two groups (Table 3).

Moreover, recurrence-free survival was also significantly longer in patients with fewer prognostic factors than in those who had more prognostic factors predicted preoperatively (Appendix A and Appendix A). The five-year recurrence-free survival was 47.4% for zero factors, 23.6% for one factor, 10.0% for two factors, and 0% for three factors. The patients without prognostic factors predicted preoperatively had fewer recurrences than those with them, and those with two or three prognostic factors had more peritoneal dissemination than those in the other two groups (Table 4).

## 3. Discussion

GBC is the most frequent biliary tract cancer according to autopsy studies; it is characterized by poor prognosis, with five-year relative survival rates ranging from 2% to 70% depending on the stage of progression. GBC has a particularly high incidence in Chile, Japan, and northern India, and is commonly associated with chronic inflammation, metaplasia progressing to dysplasia, carcinoma in situ, and then, invasive cancer. Gallstone disease, gallbladder polyps, chronic Salmonella infection, congenital biliary cysts, and abnormal pancreaticobiliary duct junctions have been reported as important precursors [7].

Factors reported to be prognostic in patients with GBC have included invasion to the bile duct [8], hepatic invasion [9], ≥4 regional lymph node metastases [10], lymph node ratio, total lymph node count [11,12], number of negative lymph nodes [13], sarcopenia [14], jaundice [15], incidental [5], volume-based PET/CT parameters of total tumor burden of malignancy [16], intraoperative bile spillage [6], T stage [17], stage [18], and resectability [19]. However, the circumstances of each research report present difficulties when attempting to generalize the results due to the limitations of each study. For example, similar to our study, Miura et al. [20] investigated the number of positive histopathological factors including direct liver invasion, HDL invasion, and lymph node metastasis; they expected good survival outcomes if there was only one factor, and poor survival if there were two or more factors. However, their study included cases of T1b, T2, and distant metastases, and it did not consider local progression factors. Their study had a sample size smaller than our study and they classified invasion to the liver and HDL differently. Chen et al. [21] found that a T stage of 3/4 was not an independent risk factor in the multivariate analysis after examining 134 patients with stage 3/4 GBC, whereas ascites, surgical margin, N stage, and pathological differentiation served as prognostic factors. However, their multivariate analysis did not examine the local progression factors of T3/4 other than liver invasion.

In comparison, our study analyzed 157 cases of only stage 3/4 GBC without distant metastases to identify prognostic factors by detailed multivariate analysis including the consideration of local progression factors. Our results and previous reports [20] suggest that even in cases of stage 3 and 4 GBC without distant metastases, hepatic invasion ≥5 mm, invasion of the left margin or entire areas of the HDL, and ≥4 regional lymph nodes metastases are strong prognostic factors, and if multiple factors are found, the R0 resection rate is low and the surgical outcome is poor. Although R0 resection did not remain a prognostic factor in this multivariate analysis, this may be because tumor characteristics such as a liver invasion ≥5 mm, ≥four regional lymph node metastases, and invasion of the left margin or the entire area of the HDL may be stronger factors than R0 resection in patients with stage 3/4 GBC.

Diagnosing the degree of progression in GBC remains problematic (Table 5). The diagnostic accuracy of multidetector computed tomography (MDCT) using combined axial and multiplanar reformation images in T-Staging was reported to be 84.9% [22]. This study found the sensitivity, specificity, and accuracy of MDCT for distinguishing GBC of ≤T2 vs. ≥T3 or ≤T3 vs. T4 in 118 patients to be 92.7%, 86.0%, and 89.8%, and 100%, 100%, and 100%, respectively [22]. The sensitivity and specificity of magnetic resonance imaging (MRI) with magnetic resonance cholangiopancreatography in detecting hepatic invasion, lymph node metastasis, and bile duct invasion are reported to be 87.5% and 86%, 60% and 90%, and 80 and 100%, respectively [23]. The correct diagnostic rate for classification of malignant hilar obstruction, which corresponds to the “invasion of the left margin or entire areas of the hepatoduodenal ligament” in our study, has been reported to be 90.5% for CT and 81.0–85.7% for MRI, and may sometimes underestimate the degree of biliary stricture [24]. Moreover, the preoperative diagnosis of lymph node metastasis in patients with GBC was also reported with a sensitivity ranging from 0.25–0.93 for CT and 0.59–0.93 for MRI, and specificity of 1.00 for CT and 0.78–1.00 for MRI [25]. Small (<10 mm) lymph node metastases were frequently undetected in pre-operative imaging [25]. Endoscopic ultrasound-guided fine-needle aspiration for diagnosis of para-aortic lymph node metastasis has a reported sensitivity of 96.7% and a specificity of 100% [26]. Therefore, there are limitations to the utility of the three factors proposed by us, especially for those with four or more lymph node metastases as preoperative factors.

One of the strengths of this study is that two of the three prognostic factors, namely hepatic invasion of 5 mm or more and bile duct stenosis indicating invasion of the left margin or the entire area of the HDL, are now being preoperatively diagnosed, to some extent, by recent advances in diagnostic imaging. Our findings imply that if both factors are found preoperatively, treatment strategies for adjuvant chemotherapy following upfront surgery, which is performed as standard treatment, can predict low R0 resection rates and extremely poor surgical outcomes. Therefore, future research should be directed at focusing on how to improve the survival rate of patients with such poor surgical prognoses, including the development of new anti-cancer drugs and molecularly targeted therapies and regimens. The development of potentially more effective anti-cancer agents, carriers for drug delivery, molecularly targeted drugs, and chemotherapy regimens followed by surgical treatment may improve outcomes for patient groups with poor prognoses, even when used in combination with current treatment strategies [29]. Furthermore, it has been reported that nutritional status is associated with the effects and outcomes of chemotherapy [30], and that inflammatory cells in the tumor microenvironment support tumor growth [31]; therefore, the maintenance of nutritional status and perioperative anti-inflammatory strategies may be important.

There were some limitations in this study. First, this single institutional retrospective study extended to three decades, during which the therapy for GBC underwent numerous changes. These changes included differences in operative approaches, diagnostic imaging [32], techniques of perioperative management, and advances in adjuvant therapy and chemotherapy (such as various chemotherapy agents), infection control, use of antimycobacterial agents, nutritional management, and progress in strategies for biliary drainage. Second, treatment strategies varied depending on the era, as mentioned in the surgical policy section in the Methods. The third major problem is that although diagnostic imaging has made great strides, accurate preoperative diagnosis of GBC remains difficult. A multicenter prospective study may be able to resolve these issues and reveal results that have fewer uncertainties.

## 4. Materials and Methods

### 4.1. Patients

This study was approved by the Institutional Review Board of the Tokyo Women’s Medical University (approval number: 4328-R2); the requirement for informed consent was waived because this was a retrospective chart review analysis.

This retrospective investigation involved 319 consecutive patients who underwent surgery for advanced GBC at our hospital from 1985 to 2017. The patients with any kind of distant metastasis (*n* = 104), stage II disease (*n* = 51), and in whom surgery for cancer was not performed (*n* = 7) were excluded. Finally, 157 patients with stage 3/4 GBC without distant metastases were included. We examined the frequency of T3 or T4 factors listed in the AJCC 8th edition, their strong prognostic factors, and postoperative outcomes depending on the number of predictable prognostic factors, preoperatively.

The median age of the patients was 69 years, and 55% were women. Two or more sectionectomies and resection of segment 4b plus 5 of the liver were performed in 29% and 27% of the patients, and simultaneous bile duct resection and PD were performed in 50% and 24% of the patients, respectively. Surgical mortality in this series was 4.5% (7/157) and details of the seven cases are described in Appendix A.

### 4.2. Surgical Policy

In our institute, hemi-hepatectomy with extrahepatic bile duct resection, hepato-pancreaticoduodenectomy (HPD), and hepato-ligament-pancreaticoduodenectomy were introduced and aggressively performed to increase curability since 1978, 1982, and 1988, respectively [19,33]. After 1997, to prevent surgical death and since the outcome was poor even after enduring surgery, we largely avoided the right hepatectomy with pancreaticoduodenectomy procedure for highly advanced GBC [34]. All the procedures involved regional lymph node dissection [3,4].

### 4.3. Postoperative Morbidity

The Clavien–Dindo Classification, which is used in various fields as a standardized classification system for assessing the severity of complications, was used to assess postoperative morbidity [35,36,37,38,39,40]. Postoperative complications are known to affect oncologic outcomes [41,42]. We classified outcomes as grade 3 or higher (where grade 3 is defined as requiring surgical, endoscopic, or radiological intervention), and grade 2 or lower (grade 2 is defined as requiring pharmacological treatment).

### 4.4. Pathological Examination

Formalin-fixed paraffin-embedded tissue sections were examined histologically, according to the AJCC cancer staging manual and the General Rules for Surgical and Pathological Studies on Cancer of the Biliary Tract of the Japanese Society of Biliary Surgery [4], and included observations regarding primary tumor status (Figure 2), lymph node involvement, and histopathological grade. The margin was evaluated by the 0 mm rule for the bile duct stump of the hepatic and duodenal sides, and the dissected periductal structure in patients who had undergone bile duct resection, or the cystic duct stump and dissected periductal structure in patients without bile duct resection. R0 was defined as negative for all stumps and no residual cancer.

Of the 79 patients who underwent bile duct resection, 41 (52%) underwent intraoperative frozen section analysis of the biliary stump, eight (8/41) of which (hepatic side: five, duodenal side: six) were initially positive. In each case, additional bile duct resection was performed; finally, a permanent pathological examination was positive in three cases (hepatic side: two, duodenal side: one). Extended surgery was withheld if cancer was suspected to remain around the dissected periductal structure or bile duct stump during surgery.

### 4.5. Adjuvant Chemotherapy

Adjuvant chemotherapy for GBC, defined as the postoperative treatment using tegafur since 1988, titanium silicate (TS)-1 since 2002, or gemcitabine since 2005 within the first 12 weeks, was administered as per the judgment of the attending physician in patients with lymph node metastasis, residual cancer, and a good postoperative condition.

### 4.6. Follow-up

The patients were monitored every 1–2 months for the first 6 months and then every 2–3 months until two years after the surgery. Recurrence was determined based on blood tests and evaluation of tumor markers (carcinoembryonic antigen and carbohydrate antigen 19-9) at every follow-up, in addition to CT or ultrasound (US) that was performed at the time of hospital consultation about every 3–6 months.

### 4.7. Statistics

Between-group differences in the qualitative and quantitative variables were determined using Fisher’s exact (two-sided) and Kruskal–Wallis rank-sum tests. The cut-off values for continuous variables were determined using the receiver operating characteristics curves analysis (Appendix A. Survival analyses were performed using the Kaplan–Meier method, log-rank test, and Cox proportional hazards model. All factors in the univariate analysis were used in the multivariate analysis. All statistical analyses were performed using R software, version 3.6.1 (R Foundation for Statistical Computing, Vienna, Austria). A *p*-value < 0.05 was considered statistically significant.

## 5. Conclusions

In conclusion, our study predicted a poor clinical outcome in patients with stage 3/4 GBC without distant metastases if two or more preoperative factors, such as hepatic invasion ≥5 mm, invasion of the left margin or the entire area of the HDL, or ≥4 lymph node metastases were recognized. Therefore, it may be necessary to develop new treatment strategies that are more expansive than surgery alone for such patients.

## Figures and Tables

**Figure 1 cancers-12-02073-f001:**
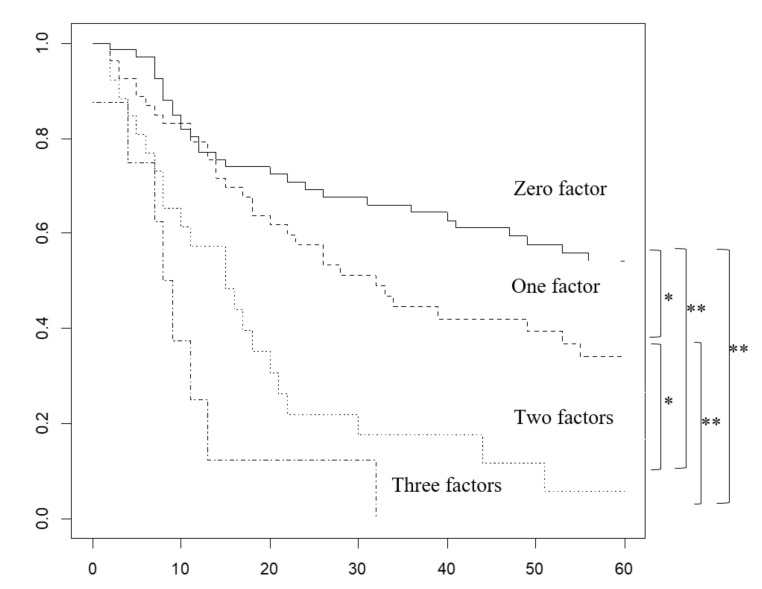
Comparison of overall survival (OS) based on the number of prognostic factors predicted preoperatively in patients with stage 3/4 gallbladder cancer without distant metastases. * < 0.05, ** < 0.01.

**Figure 2 cancers-12-02073-f002:**
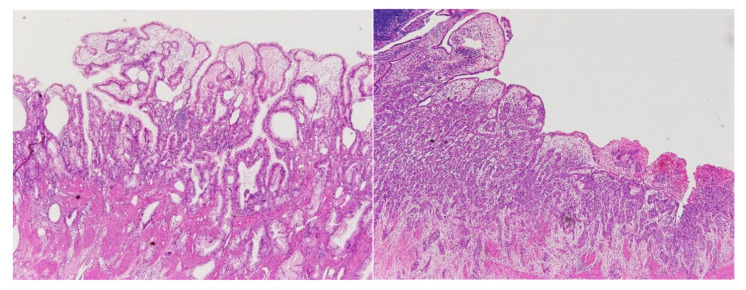
Histological images (hematoxylin and eosin stain, 4× objective lens) of tubular adenocarcinomas representing the moderately-differentiated type (left) and the poorly-differentiated type (right).

**Table 1 cancers-12-02073-t001:** Frequency and R0 resection rate by local factors of primary site in patients with stage 3/4 GBC (*n* = 157) without distant metastasis.

Tumor Factors	Incidence	R0 Rate
Tumor perforates the serosa or more	61% (95)	76% (72/95)
Direct invasion to the liver parenchyma, <5 mm	17% (27)	93% (25/27)
Direct invasion to the liver parenchyma, ≥5 mm, <20 mm	17% (26)	65% (17/26)
Direct invasion to the liver parenchyma, ≥20 mm	18% (28)	82% (23/28)
Invasion to the right margin, but not to the left margin of the HDL	15% (24)	83% (20/24)
Invasion to the left margin, but not to the entire HDL	24% (37)	63% (23/37)
Invasion through the HDL	10% (16)	56% (9/16)
Stomach or Duodenum invasion	11% (17)	65% (11/17)
Colon invasion	16 (10%)	69% (11/16)
Pancreas invasion	0.6% (1)	0% (0/1)
Two or more extrahepatic organs/structures	13% (20)	55% (11/20)
Main portal vein or hepatic artery	24% (37)	59% (22/37)
AJCC 8th 1–3 regional LNM	59% (92)	84% (77/92)
AJCC 8th ≥4 regional LNM	16% (25)	64% (16/25)

Numbers in parentheses indicate the number of cases. HDL, hepatoduodenal ligament: AJCC 8th, the 8th American Joint Committee Classification; LNM, lymph node metastasis.

**Table 2 cancers-12-02073-t002:** Univariate and multivariate analyses of overall survival in patients with resected stage III-IV gallbladder cancer without distant metastasis.

Factors	^a^ Univariate			^b^ Multivariate	
	*n*	5y OS %	*p* Value	HR (95% CI)	*p* Value
Period 2000–2017 (to 1985–1999)	93/64	35/41	1.0	1.62 (0.90–2.92)	0.11
Age ≥67 (to <67)	92/65	33/43	0.30	1.14 (0.69–1.90)	0.60
Sex (female to male)	86/71	39/35	0.80	0.87 (0.53–1.43)	0.58
Incidental (to none)	16/141	38/37	0.90	1.40 (0.63–3.12)	0.41
Cholecystectomy (to LBR)	33/36	43/46	0.60	1.65 (0.75–3.60)	0.13
S4bS5 (to LBR)	42/36	38/46	0.40	0.49 (0.21–1.15)	0.10
Hepatectomy ≥2 sections (to LBR)	46/36	24/46	0.03	0.56 (0.24–1.35)	0.20
Without BDR (to with BDR)	40/79	50/26	0.03	0.86 (0.35–2.09)	0.73
Pancreatoduodenectomy (to BDR)	38/79	45/26	0.10	0.67 (0.34–1.34)	0.26
Blood loss ≥1400 g (to <1400 g)	60/97	26/43	0.003	2.19 (1.26–3.80)	0.005
Surgery time ≥258 min (to <258 min)	113/44	32/50	0.03	1.23 (0.55–2.73)	0.61
Histology papillary (to well/moderate)	26/79	56/34	0.20	1.66 (0.71–3.89)	0.24
Histology poorly/others (to well/moderate)	51/79	29/34	0.05	2.26 (1.33–3.84)	0.003
Liver invasion <5 mm (to no invasion)	27/76	40/48	0.50	1.24 (0.56–2.72)	0.59
Liver invasion ≥5 mm (to no invasion)	54/76	19/48	<0.001	2.42 (1.17–5.01)	0.017
Binf1 (to no invasion)	24/80	41/49	0.30	0.73 (0.31–1.70)	0.47
Binf2/3 (to no invasion)	53/80	17/49	<0.001	2.10 (0.99–4.46)	0.053
≥2 organs or structure invasions (to <1)	20/137	21/40	0.02	1.05 (0.51–2.20)	0.89
MPV or HA invasions (to no invasions)	35/119	18/43	<0.001	1.75 (0.87–3.51)	0.11
AJCC8th 1–3 regional LNM (to N0)	92/40	39/47	0.50	1.34 (0.74–2.39)	0.33
AJCC8th ≥4 regional LNM (to N0)	25/40	11/47	0.001	2.25 (1.07–4.74)	0.034
Residual cancer R1.2 (to R0)	32/125	14/42	0.004	1.09 (0.55–2.16)	0.81
Clavien–Dindo classification ≥3 (to ≤2)	65/92	29/43	0.02	1.64 (0.99–2.71)	0.054
Without adjuvant chemotherapy (to with)	117/40	36/41	0.20	1.93 (1.03–3.62)	0.041

^a^ Log-rank test, ^b^ Cox proportional hazards model. OS, overall survival; HR, hazard ratio; CI, confidence interval; S4bS5, resection of segment 4b plus 5 of the liver; LBR, liver bed resection; BDR, Bile duct resection; HDL, hepatoduodenal ligament; Binf1, invasion of the right margin of the HDL, but not to the left margin; Binf2/3, invasion of the left margin of the HDL, but not to the entire ligament/entire area of the HDL; MPV, main portal vein; HA, hepatic artery; AJCC, American Joint Committee on Cancer; LNM, lymph node metastasis; N0, no lymph node metastasis; R0, curative margin-free resection; R1.2, positive margin resection with or without any other residual cancer.

**Table 3 cancers-12-02073-t003:** Comparison of characteristics among groups according to the number of prognostic factors in patients with stage 3/4 gallbladder cancer without distant metastases.

Factors	0 Factors	1 Factor	2 or 3 Factors	*p* Value ^Φ^
	(*n* = 68)	(*n* = 54)	(*n* = 35)	
Period 2000-17	38 (56%)	34 (63%)	21 (60%)	0.73
Age (y, median)	71	69	66	0.20 ^ε^
Female	39 (57%)	25 (46%)	22 (63%)	0.26
Incidental	10 (15%)	4 (7.4%)	2 (5.7%)	0.32
Liver invasion ≥5 mm	0 #, *	24 (44%) #, ^∀^	30 (86%) *,^∀^	<0.001
Invasion of the left margin, or entire area of the HDL	0 #, *	24 (44%) #, ^∀^	29 (83%) *, ^∀^	<0.001
≥4 regional lymph node metastasis	0 #, *	6 (11%) #, ^∀^	19 (54%) *, ^∀^	<0.001
Hepatectomy (LB/GB/S45/≥2 sections)	27/25/15/1 #, *	6/6/17/15 #	3/2/10/20 *	<0.001
BDR (with/without/PD)	22/29/17 #, *	34/10/10 #	23/1/11 *	<0.001
Vascular resection	1 (1.5%) #, *	6 (11%) #, ^∀^	17 (49%) *, ^∀^	<0.001
Blood loss (g)	765 #, *	1025 #	1498 *	<0.001 #, ^ε^
Surgery time (min)	261 #, *	370 #	400 *	<0.001 ^ε^
Histology (tub1 or 2/pap/poor or others)	30/22/16 #, *	28/3/23 #	21/1/12 *	<0.001
≥2 organs or structure invasions	1 (1.5%) #, *	6 (11%) #, ^∀^	13 (37%) *, ^∀^	<0.001
MPV or HA invasions	2 (2.9%) #, *	12 (22%) #, ^∀^	23 (66%) *, ^∀^	<0.001
R0	60 (88%) #	48 (89%) *	18 (51%) #, *	<0.001
Clavien–Dindo classification ≥3	20 (29%) #	25 (46%) *	20 (57%) #, *	0.018
Adjuvant chemotherapy	17 (25%)	14 (26%)	9 (26%)	1.0

# *p* < 0.05, between 0 vs. 1 factor; * *p* < 0.05, between 0 vs. 2 or 3 factors; ^∀^
*p* < 0.05 between 1 vs. 2 or 3 factors with Fisher exact test two sided, respectively. ^Φ^ 2 × 3 Fisher exact test two sided, ^ε^ Kruskal–Wallis rank sum test. HDL, hepatoduodenal ligament; LB, liver bed resection; GB, cholecystectomy; S45, resection of segment 4a plus 5 of the liver; BDR, bile duct resection; PD, pancreaticoduodenectomy; MPV, main portal vein; HA, hepatic artery; R0, curative margin-free resection.

**Table 4 cancers-12-02073-t004:** Comparison of recurrence patterns among groups by number of preoperative predictable prognostic factors in patients with stage 3/4 gallbladder cancer without distant metastases.

Type of Recurrence	0 Factors	1 Factor	2 or 3 Factors	*p* Value ^Φ^
	(*n* = 68)	(*n* = 54)	(*n* = 35)	
Any type of recurrence	32 (47%) #, *	38 (70%) #	26 (74%) *	0.007
Liver	11 (16%) #	17/52 (33%) #	6/33 (18%)	0.09
Lymph node	9 (16%)	13/52 (25%)	5/33 (15%)	0.23
Dissemination	10 (15%) *	10/52 (19%) ^∀^	13/33 (39%) *, ^∀^	0.02
Local	9 (16%)	12/52 (23%)	8/33 (24%)	0.61
Others	1	4 (44%)	3 (83%)	0.11

# *p* < 0.05, between 0 vs. 1 factor; * *p* < 0.05, between 0 vs. 2 or 3 factors; ^∀^*p* < 0.05 between 1 vs. 2 or 3 factors with Fisher exact test two sided, respectively. ^Φ^ 2 × 3 Fisher exact test two sided. Preoperatively predictable prognostic factors are defined as liver invasion ≥5 mm, invasion of the left margin, or entire area of the HDL and ≥4 regional lymph node metastases.

**Table 5 cancers-12-02073-t005:** Reported sensitivity, specificity, and accuracy of diagnosis in patients with gallbladder cancer.

Diagnosed Factors	Sensitivity		Specificity		Accuracy	
	MRI	CT	MRI	CT	MRI	CT
≤T1a vs≥T1b [27,28]	82.7–89.3	NA	100	NA	93.0–95.4	NA
≤T1vs≥T2 [27,28]	88.1–94.0	79.3	79.0–84.2	98.8	87.2–90.7	94.7
≤T2vs≥T3 [27,28]	83.3–90.0	92.7	80.4–89.4	86	80.2–88.4	89.3
≤T3vs≥T4 [27,28]	100	100	59.5–100	100	96.5–100	100
Liver invasion [23,24]	87.5	NA	86	NA	NA	NA
Biliary invasion [23,24]	80	NA	100	NA	81–85.7	90.5
Lymph node Mets [23,24,25,27]	25–71	81.8–93.0	42–100	70–100	NA	NA
Hepatic artery invasion [25]	50–66.7	83.3	53.3–73.3	93.3–100	NA	NA
Portal vein invasion [24]	80	80–90	72.7–81.8	90.9	NA	NA
Duodenal invasion [23]	50	NA	100	NA	NA	NA
Liver or peritoneal Mets	93.8	80–100	93.8–100	80	NA	NA

Mets, metastasis; NA, not available; T1a, Tumor invades the laminar propria; T1b, Tumor invades the muscular layer; T1, Tumor invades the laminar propria or muscular layer; T2, Tumor invades the perimuscular connective tissue; T3, Tumor perforates the serosa and/or directly invades the liver and/or one other adjacent organ or structure such as the stomach, duodenum, colon, pancreas, omentum, or extrahepatic bile ducts; T4, Tumor invades the main portal vein or hepatic artery or invades two or more extrahepatic organs or structures; MRI, magnetic resonance imaging; CT, computed tomography.

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
