# Peer review of "Examination of Prognostic Factors Affecting Long-Term Survival of Patients with Stage 3/4 Gallbladder Cancer without Distant Metastasis"

_cancers, 2020, doi:10.3390/cancers12082073_

Round 1

Reviewer 1 Report

English (grammar and proofing) required.

L19: it would be helpful to detail some of the”multiple prognostic factors” in the abstract

L23: was the surgical mortality 4.5% for the 157 cases detailed in the abstract? This is not the case in the main text.

L81: The journal is aimed at a broad audience of cancer biologists through to medical professionals, not just surgeons, therefore could the authors attempt to make the article more accessible to this audience? For example more detail on Clavien-Dindo classification for non-medics and how this links to reliable measures of outcomes.

L97: It is more customary to split the dotted line into two, one line for two factors and one for three. What is the classification accuracy therein? Is it appropriate to show a ROC approach?

L151: The sensitivity and specificity and accuracy of MDCT for GBC of different T are unclear; a number of percentages are quoted but represent a colloquium of data. Can the authors please summarise this in a table quoting the sens, specif and acc for each of the groups.

L179: The authors describe the limitations of the study concisely however I am concerned that these are not described (a) in enough detail, and (b) as objectively as possible. Indeed, a recent review carried out in a bespoke surgical journal saw a specific sections on limitations,  with >3/4 page dedicated to this. Whilst this will not detract from the impact of this well-written article, it is imperative that medical reports such as that described herein are accompanied by objective and rationalised limitation studies. Please therefore expand this section in line with these comments.

L179: Please also offer further perspectives for the future and future work: based on this substantial study, there is a unique opportunity to expand the paper in order to provide a meaningful and impactful commentary for future perspectives and work going forward.

L194: Can the authors please describe the nature of 4.5% surgical mortalities and their underlying the stats?

Additional references pertinent to this manuscript requiring insertion:

Kubyshkin, V. and Budisa, N., 2017. Amide rotation trajectories probed by symmetry. Organic & biomolecular chemistry15(32), pp.6764-6772.

Clavien, P.A., Barkun, J., De Oliveira, M.L., Vauthey, J.N., Dindo, D., Schulick, R.D., De Santibañes, E., Pekolj, J., Slankamenac, K., Bassi, C. and Graf, R., 2009. The Clavien-Dindo classification of surgical complications: five-year experience. Annals of surgery250(2), pp.187-196.

Lee, J.H., Park, D.J., Kim, H.H., Lee, H.J. and Yang, H.K., 2012. Comparison of complications after laparoscopy-assisted distal gastrectomy and open distal gastrectomy for gastric cancer using the Clavien–Dindo classification. Surgical endoscopy26(5), pp.1287-1295.

Grootveld, M., Percival, B., Gibson, M., Osman, Y., Edgar, M., Molinari, M., Mather, M.L., Casanova, F. and Wilson, P.B., 2019. Progress in low-field benchtop NMR spectroscopy in chemical and biochemical analysis. Analytica chimica acta, 1067, pp.11-30.

Lacy, A.M., Tasende, M.M., Delgado, S., Fernandez-Hevia, M., Jimenez, M., De Lacy, B., Castells, A., Bravo, R., Wexner, S.D. and Heald, R.J., 2015. Transanal total mesorectal excision for rectal cancer: outcomes after 140 patients. Journal of the American College of Surgeons221(2), pp.415-423.

Tokunaga, M., Kondo, J., Tanizawa, Y., Bando, E., Kawamura, T. and Terashima, M., 2012. Postoperative intra-abdominal complications assessed by the Clavien–Dindo classification following open and laparoscopy-assisted distal gastrectomy for early gastric cancer. Journal of Gastrointestinal Surgery16(10), pp.1854-1859.

Owen, L., Laird, K. and Wilson, P.B., 2018. Structure-activity modelling of essential oils, their components, and key molecular parameters and descriptors. Molecular and cellular probes38, pp.25-30.

Panhofer, P., Ferenc, V., Schütz, M., Gleiss, A., Dubsky, P., Jakesz, R., Gnant, M. and Fitzal, F., 2014. Standardization of morbidity assessment in breast cancer surgery using the Clavien Dindo Classification. International Journal of Surgery12(4), pp.334-339.

Uehara, T., Hirode, M., Ono, A., Kiyosawa, N., Omura, K., Shimizu, T., Mizukawa, Y., Miyagishima, T., Nagao, T. and Urushidani, T., 2008. A toxicogenomics approach for early assessment of potential non-genotoxic hepatocarcinogenicity of chemicals in rats. Toxicology250(1), pp.15-26.

Leenders, J., Grootveld, M., Percival, B., Gibson, M., Casanova, F. and Wilson, P.B., 2020. Benchtop Low-Frequency 60 MHz NMR Analysis of Urine: A Comparative Metabolomics Investigation. Metabolites10(4), p.155.

Alyassin, Y., Sayed, E.G., Mehta, P., Ruparelia, K., Arshad, M.S., Rasekh, M., Shepherd, J., Kucuk, I., Wilson, P.B., Singh, N. and Chang, M.W., 2020. Application of mesoporous silica nanoparticles as drug delivery carriers for chemotherapeutic agents. Drug Discovery Today.

Radu, V., Price, J.C., Levett, S.J., Narayanasamy, K.K., Bateman-Price, T.D., Wilson, P.B. and Mather, M.L., 2019. Dynamic quantum sensing of paramagnetic species using nitrogen-vacancy centers in diamond. ACS sensors5(3), pp.703-710.

Duraes, L.C., Stocchi, L., Steele, S.R., Kalady, M.F., Church, J.M., Gorgun, E., Liska, D., Kessler, H., Lavryk, O.A. and Delaney, C.P., 2018. The relationship between Clavien–Dindo morbidity classification and oncologic outcomes after colorectal cancer resection. Annals of surgical oncology25(1), pp.188-196.

Jiang, X., Hiki, N., Nunobe, S., Fukunaga, T., Kumagai, K., Nohara, K., Sano, T. and Yamaguchi, T., 2011. Postoperative outcomes and complications after laparoscopy-assisted pylorus-preserving gastrectomy for early gastric cancer. Annals of surgery253(5), pp.928-933.

Percival, B.C., Gibson, M., Wilson, P.B., Platt, F.M. and Grootveld, M., 2020. Metabolomic Studies of Lipid Storage Disorders, with Special Reference to Niemann-Pick Type C Disease: A Critical Review with Future Perspectives. International Journal of Molecular Sciences21(7), p.2533.

McSorley, S.T., Watt, D.G., Horgan, P.G. and McMillan, D.C., 2016. Postoperative systemic inflammatory response, complication severity, and survival following surgery for colorectal cancer. Annals of surgical oncology23(9), pp.2832-2840.

Author Response

We thank the editor and the reviewers for the thorough assessment of our manuscript. We have provided point-by-point responses to each of the comments beginning on the following page, and the revised portions of our manuscript are shown in colored text. We hope the changes made to our manuscript have made it more suitable for publication in your journal. We thank you for your consideration and look forward to hearing from you.

Reviewer 1

English (grammar and proofing) required.

Thank you for your comment. The language and grammar have been reviewed by the English-language professional editing service.

L19: it would be helpful to detail some of the”multiple prognostic factors” in the abstract

We explain “multiple prognostic factors” in the Abstract as follows:

Abstract

L23: In multivariate analysis, blood loss, poor histology, liver invasion, and ≥4 regional lymph node metastases (LNM) were independent prognostic factors for poor surgical outcomes; invasion of the left margin or the entire area of the hepatoduodenal ligament and a Clavien-Dindo classification ≥3 were marginal factors. The analysis identified outcomes of patients with factors that could be predicted preoperatively, such as liver invasion ≥5 mm, invasion of the left margin or the entire area of the hepatoduodenal ligament, and ≥4 regional LNMs.

L23: was the surgical mortality 4.5% for the 157 cases detailed in the abstract? This is not the case in the main text.

Thank you for your comment. We have revised the sentence as indicated below, and we describe the details of the seven mortality cases in Supplementary Table 3:

L220: Surgical mortality in this series was 4.5% (7/157) and details of these seven cases are described in Supplementary Table 3.

L81: The journal is aimed at a broad audience of cancer biologists through to medical professionals, not just surgeons, therefore could the authors attempt to make the article more accessible to this audience? For example more detail on Clavien-Dindo classification for non-medics and how this links to reliable measures of outcomes.

Thank you for your comments. I have added an explanation of the Clavien-Dindo Classification so that non-medical personnel can better understand it in the Methods section, as quoted below:

L230:

 4.3. Postoperative morbidity

 The Clavien-Dindo Classification, which is used in various fields as a standardized classification system for assessing the severity of complications, was used to assess postoperative morbidity [34-39]. Postoperative complications are known to affect oncologic outcomes [40,41]. We classified outcomes as grade 3 or higher (grade 3 is defined as requiring surgical, endoscopic, or radiological intervention), and grade 2 or lower (where grade 2 is defined as requiring pharmacological treatment).

L97: It is more customary to split the dotted line into two, one line for two factors and one for three. What is the classification accuracy therein? Is it appropriate to show a ROC approach?

Thank you for your comment. According to your suggestion, we have described two factors and three factors separately in the figure. This is a classification based on pathological diagnoses, so there is no data about its accuracy.

L94: Liver invasion of 5 mm or more, left marginal invasion or invasion of the entire HDL, and metastases of four or more regional lymph node were considered predictable prognostic factors, to some extent, before surgery (Figure 1 and Table S2A). The 5-year survival rate was 54.2% for 0 factors, 34.2% for 1 factor, 5.87% for 2 factors, and 0% for 3 factors.

L151: The sensitivity and specificity and accuracy of MDCT for GBC of different T are unclear; a number of percentages are quoted but represent a colloquium of data. Can the authors please summarise this in a table quoting the sens, specif and acc for each of the groups.

We have created and inserted a table (Table 5) according to your recommendations.

L176:

Table 5. Reported sensitivity, specificity, and accuracy of diagnosis in patients with gallbladder cancer.

Sensitivity

Specificity

accuracy

MRI

CT

MRI

CT

MRI

CT

≤T1a vs≥T1b [27,28]

82.7-89.3

NA

100

NA

93.0-95.4

≤T1vs≥T2 [27,28]

88.1-94.0

79.3

79.0-84.2

98.8

87.2-90.7

94.7

≤T2vs≥T3 [27,28]

83.3-90.0

92.7

80.4-89.4

86

80.2-88.4

89.3

≤T3vs≥T4 [27,28]

100

100

59.5-100

100

96.5-100

100

Liver invasion [23,24]

87.5

NA

86

NA

NA

NA

Biliary invasion  [23,24]

80

NA

100

NA

81-85.7

90.5

Lymph node mets [23-25,27]

25-71

81.8-93.0

42-100

70-100

NA

NA

Hepatic artery invasion [25]

50-66.7

83.3

53.3-73.3

93.3-100

NA

NA

Portal vein invasion [24]

80

80-90

72.7-81.8

90.9

Duodenal invasion [23]

50

NA

100

NA

NA

NA

Liver or peritoneal mets

93.8

80-100

93.8-100

80

NA

NA

Mets, metastasis; NA, not available; T1a, Tumor invades the laminar propria; T1b, Tumor invades the muscular layer; T1, Tumor invades the laminar propria or muscular layer; T2, Tumor invades the perimuscular connective tissue; T3, Tumor perforates the serosa and/or directly invades the liver and/or one other adjacent organ or structure such as the stomach, duodenum, colon, pancreas, omentum, or extrahepatic bile ducts; T4, Tumor invades the main portal vein or hepatic artery or invades two or more extrahepatic organs or structures; MRI, magnetic resonance imaging; CT, computed tomography.

L179: The authors describe the limitations of the study concisely however I am concerned that these are not described (a) in enough detail, and (b) as objectively as possible. Indeed, a recent review carried out in a bespoke surgical journal saw a specific sections on limitations,  with >3/4 page dedicated to this. Whilst this will not detract from the impact of this well-written article, it is imperative that medical reports such as that described herein are accompanied by objective and rationalised limitation studies. Please therefore expand this section in line with these comments.

Thank you for your comments. We have modified the sentences about the study’s limitations as follows:

L194: There were some limitations in this study. First, this single institutional retrospective study extended to three decades, during which the therapy for GBC underwent numerous changes. These changes included differences in operative approaches, diagnostic imaging, techniques of perioperative management, and advances in adjuvant therapy and chemotherapy (such as various chemotherapy agents), infection control, use of antimycobacterial agents, nutritional management, and progress in strategies for biliary drainage. Second, treatment strategies varied depending on the era, as mentioned in the surgical policy section of the Methods. The third major problem is that although diagnostic imaging has made great strides, accurate preoperative diagnosis of GBC remains difficult. A multicenter prospective study may be able to resolve these issues and reveal results that have fewer uncertainties.

L179: Please also offer further perspectives for the future and future work: based on this substantial study, there is a unique opportunity to expand the paper in order to provide a meaningful and impactful commentary for future perspectives and work going forward.

 Thank you for your comments. We have added the sentence below:

L187: Development of potentially more effective anti-cancer agents, drug carriers, molecularly targeted drugs, and chemotherapy regimens followed by surgical treatment may improve outcomes for patient groups with poor prognoses, even when used in combination with current treatment strategies [29]. Furthermore, it has been reported that nutritional status is associated with the effects and outcomes of chemotherapy [30], and that inflammatory cells in the tumor microenvironment support tumor growth [31]; therefore maintenance of nutritional status and perioperative anti-inflammatory strategies may be important.

L194: Can the authors please describe the nature of 4.5% surgical mortalities and their underlying the stats?

Thank you for your comment.

 We have added the denominator and numerator of the surgical mortality rate, and have detailed this information in the supplementary table 3.

Additional references pertinent to this manuscript requiring insertion:

Thank you for your suggestion; we have assessed each reference individually for its suitability for inclusion: 

Kubyshkin, V. and Budisa, N., 2017. Amide rotation trajectories probed by symmetry. Organic & biomolecular chemistry15(32), pp.6764-6772.

I could not cite this study which amide rotation of peptidyl–prolyl fragments trajectories probed by symmetry, as it has little relation to our own. 

Clavien, P.A., Barkun, J., De Oliveira, M.L., Vauthey, J.N., Dindo, D., Schulick, R.D., De Santibañes, E., Pekolj, J., Slankamenac, K., Bassi, C. and Graf, R., 2009. The Clavien-Dindo classification of surgical complications: five-year experience. Annals of surgery250(2), pp.187-196.

I have included a citation of the study by Clavien et al. 

Lee, J.H., Park, D.J., Kim, H.H., Lee, H.J. and Yang, H.K., 2012. Comparison of complications after laparoscopy-assisted distal gastrectomy and open distal gastrectomy for gastric cancer using the Clavien–Dindo classification. Surgical endoscopy26(5), pp.1287-1295.

I have included a citation of the study by Lee et al. 

 Grootveld, M., Percival, B., Gibson, M., Osman, Y., Edgar, M., Molinari, M., Mather, M.L., Casanova, F. and Wilson, P.B., 2019. Progress in low-field benchtop NMR spectroscopy in chemical and biochemical analysis. Analytica chimica acta, 1067, pp.11-30.

I have included a citation of the study by Grootveld et al.   

Lacy, A.M., Tasende, M.M., Delgado, S., Fernandez-Hevia, M., Jimenez, M., De Lacy, B., Castells, A., Bravo, R., Wexner, S.D. and Heald, R.J., 2015. Transanal total mesorectal excision for rectal cancer: outcomes after 140 patients. Journal of the American College of Surgeons221(2), pp.415-423.

I have included a citation of the study by Lacy et al.  

Tokunaga, M., Kondo, J., Tanizawa, Y., Bando, E., Kawamura, T. and Terashima, M., 2012. Postoperative intra-abdominal complications assessed by the Clavien–Dindo classification following open and laparoscopy-assisted distal gastrectomy for early gastric cancer. Journal of Gastrointestinal Surgery16(10), pp.1854-1859.

I have included a citation of the study by Tokunaga et al.  

Owen, L., Laird, K. and Wilson, P.B., 2018. Structure-activity modelling of essential oils, their components, and key molecular parameters and descriptors. Molecular and cellular probes38, pp.25-30.

I could not cite this study which Structure-activity modelling of essential oils, their components, and key molecular parameters and descriptors, it has little relation to our own.   

Panhofer, P., Ferenc, V., Schütz, M., Gleiss, A., Dubsky, P., Jakesz, R., Gnant, M. and Fitzal, F., 2014. Standardization of morbidity assessment in breast cancer surgery using the Clavien Dindo Classification. International Journal of Surgery12(4), pp.334-339.

I have included a citation of the study by Panhofer et al.   

Uehara, T., Hirode, M., Ono, A., Kiyosawa, N., Omura, K., Shimizu, T., Mizukawa, Y., Miyagishima, T., Nagao, T. and Urushidani, T., 2008. A toxicogenomics approach for early assessment of potential non-genotoxic hepatocarcinogenicity of chemicals in rats. Toxicology250(1), pp.15-26.

I could not cite this study which A toxicogenomics approach for early assessment of potential non-genotoxic hepatocarcinogenicity of chemicals in rats, it has little relation to our own.   

Leenders, J., Grootveld, M., Percival, B., Gibson, M., Casanova, F. and Wilson, P.B., 2020. Benchtop Low-Frequency 60 MHz NMR Analysis of Urine: A Comparative Metabolomics Investigation. Metabolites10(4), p.155.

I could not cite this study which Benchtop Low-Frequency 60 MHz NMR Analysis of Urine: A Comparative Metabolomics Investigation, it has little relation to our own.  

Alyassin, Y., Sayed, E.G., Mehta, P., Ruparelia, K., Arshad, M.S., Rasekh, M., Shepherd, J., Kucuk, I., Wilson, P.B., Singh, N. and Chang, M.W., 2020. Application of mesoporous silica nanoparticles as drug delivery carriers for chemotherapeutic agents. Drug Discovery Today.

I have included a citation of the study by Alyassin et al. 

Radu, V., Price, J.C., Levett, S.J., Narayanasamy, K.K., Bateman-Price, T.D., Wilson, P.B. and Mather, M.L., 2019. Dynamic quantum sensing of paramagnetic species using nitrogen-vacancy centers in diamond. ACS sensors5(3), pp.703-710.

I could not cite this study which Dynamic Quantum Sensing of Paramagnetic Species Using Nitrogen-Vacancy Centers in Diamond, it has little relation to our own.   

Duraes, L.C., Stocchi, L., Steele, S.R., Kalady, M.F., Church, J.M., Gorgun, E., Liska, D., Kessler, H., Lavryk, O.A. and Delaney, C.P., 2018. The relationship between Clavien–Dindo morbidity classification and oncologic outcomes after colorectal cancer resection. Annals of surgical oncology25(1), pp.188-196.

I have included a citation of the study by Duraes et al.  

Jiang, X., Hiki, N., Nunobe, S., Fukunaga, T., Kumagai, K., Nohara, K., Sano, T. and Yamaguchi, T., 2011. Postoperative outcomes and complications after laparoscopy-assisted pylorus-preserving gastrectomy for early gastric cancer. Annals of surgery253(5), pp.928-933.

 I have included a citation of the study by Jiang et al. 

Percival, B.C., Gibson, M., Wilson, P.B., Platt, F.M. and Grootveld, M., 2020. Metabolomic Studies of Lipid Storage Disorders, with Special Reference to Niemann-Pick Type C Disease: A Critical Review with Future Perspectives. International Journal of Molecular Sciences21(7), p.2533.

I could not cite this study which DMetabolomic Studies of Lipid Storage Disorders, with Special Reference to Niemann-Pick Type C Disease: A Critical Review with Future Perspectives, it has little relation to our own.  

 McSorley, S.T., Watt, D.G., Horgan, P.G. and McMillan, D.C., 2016. Postoperative systemic inflammatory response, complication severity, and survival following surgery for colorectal cancer. Annals of surgical oncology23(9), pp.2832-2840.

I have included a citation of the study by McSorley et al.

Reviewer 2 Report

The authors presented a vey interesting paper looking for the prognostic factors in stage 3/4 gallbladder cancer patients. The cohort of patients seems to be important and both pathological and clinical data were reported. The AJCC classification of surgical specimens is well addressed according the 8th edition of AJCC cancer staging manual. Indeed, some information might be provided from the authors that could increase the soundness of manuscript.

So far, we would like to suggest to the authors the following list of observations.

Minor Comments:

1) The authors described very well the pathological data of patients. However, the description of pathological procedure in order to perform the R margins analyses are not very well reported (i.e. criteria of distance, how the pathologist evaluate this aspects, ect..).

2) The authors performed DBR surgical procedure in 79 cases (if we interpretated the data, correctly). Did the Authors performed Surgical Margin examination during the BDR procedure? If yes, How many time? Furthermore, Ddid these results changed the surgical procedure established at the beginning?

3) The R0 value seems to be important in the results and evaluation of additional parameters. Looking in the univariate and multivariate analyses reported in table 2, the p values of R0 seem to be very distant. Have the authors additional comments to do?

4) Have the authors a good number of patient in order to compare Stage IVA vs Stage IVB?

5) We would like to suggest to the authors to add some histological images in order to represent better the GBC.

Author Response

We thank the editor and the reviewers for the thorough assessment of our manuscript. We have provided point-by-point responses to each of the comments beginning on the following page, and the revised portions of our manuscript are shown in colored text. We hope the changes made to our manuscript have made it more suitable for publication in your journal. We thank you for your consideration and look forward to hearing from you.

Reviewer 2

The authors presented a very interesting paper looking for the prognostic factors in stage 3/4 gallbladder cancer patients. The cohort of patients seems to be important and both pathological and clinical data were reported. The AJCC classification of surgical specimens is well addressed according the 8th edition of AJCC cancer staging manual. Indeed, some information might be provided from the authors that could increase the soundness of manuscript.

So far, we would like to suggest to the authors the following list of observations.

Minor Comments:

  • The authors described very well the pathological data of patients. However, the description of pathological procedure in order to perform the R margins analyses are not very well reported (i.e. criteria of distance, how the pathologist evaluate this aspects, ect..).

Thank you for your valuable comment. We have added the information as indicated below:

L249: The margin was evaluated by the 0 mm rule for the bile duct stump of the hepatic and duodenal sides, and the dissected periductal structure in patients who had undergone bile duct resection, or the cystic duct stump and dissected periductal structure in patients without bile duct resection. R0 was defined as negative for all stumps and no residual cancer.

  • The authors performed DBR surgical procedure in 79 cases (if we interpretated the data, correctly). Did the Authors performed Surgical Margin examination during the BDR procedure? If yes, How many time? Furthermore, Ddid these results changed the surgical procedure established at the beginning?

Thank you for your comment. We have added the information indicated below:

L253: Of the 79 patients who underwent bile duct resection, 41 (52%) underwent intraoperative frozen section analysis of the biliary stump, eight (8/41) of which (hepatic side: five, duodenal side: six) were initially positive. In each case, additional bile duct resection was performed; finally, a permanent pathological examination was positive in three cases (hepatic side: two, duodenal side: one). Extended surgery was withheld if cancer was suspected to remain around the dissected periductal structure or bile duct stump during surgery.

  • The R0 value seems to be important in the results and evaluation of additional parameters. Looking in the univariate and multivariate analyses reported in table 2, the p values of R0 seem to be very distant. Have the authors additional comments to do?

Thank you for your comment. We have added the following sentence to the Discussion section:

L153: Although R0 resection did not remain a prognostic factor in this multivariate analysis, this may be because tumor characteristics such as liver invasion ≥ 5 mm, ≥ four regional lymph node metastases, and invasion of the left margin or the entire area of the hepatoduodenal ligament may be stronger factors than R0 resection in patients with stage 3/4 GBC.

  • Have the authors a good number of patients in order to compare Stage IVA vs Stage IVB?

Thank you for your comment. Thirty-one cases of stage IVA and 25 cases of stage IVB GBCs were included in the study.

We have added the following information to Supplementary Table 3 showing the numbers and percentages of patients at each stage:

AJCC stage (IIIA/IIIB/IVA/IVB)   33 (21%) /68 (43%) /31(20%) /25 (16%)

5) We would like to suggest to the authors to add some histological images in order to represent better the GBC.

Thank you for your valuable comment.

We have added Figure 2, which contains histological images of tubular adenocarcinomas representing the moderately-differentiated type (left) and the poorly-differentiated type (right).

Reviewer 3 Report

The manuscript entitled “Examination of Prognostic Factors Affecting Long term Survival of Patients with Stage 3/4 Gallbladder Cancer without Distant Metastasis” is an intrigue original retrospective study improving clinical practice in gallbladder cancer.

The Authors focus their attention on the role prognostic/predictive biomarkers for precision medicine in biliary cancers.

This paper is well written and balance, so far it should be published after some minor revisions: 

  • abbreviation should be checked
  • please revised the mistyping
  • English should be revised
  • In univariate and multivariate analysis, Authors consider several factors. In any case, only few of them result statistically significant. Later, in Overall survival rate based on the number of prognostic factors predicted preoperative paragraph, the Authors write Further, the 5-year survival rate was 54.2% for 0 factors, 34.2% for 1 factor, and 4.42% for 2 or 3 94 factors, when liver invasion was 5 mm or more, left marginal invasion or the entire HDL, and 95 metastases of 4 or more regional lymph nodes were considered as predictable prognostic factors to 96 some extent before surgery (Figure 1 and Table S2A). The factors are not the same. Please specify.
  • Figure 2 should be improved (it contains overlapping images)
  • Discussion: Authors should write a brief introduction on clinical/biological features of CCA. Please better the introduction with the informations required (they should refer to Current opinion on clinical practice diagnostic and therapeutic algorithms: A review of the literature and a long-standing experience of a referral center. Dig Liver Dis. 2016 Mar;48(3):231-41)
  • It might be interesting to understand the possibility of recovering data on the nutritional or inflammatory status of these patients, since they could represent prognostic factors for this pathology (Authors can refer to: a) The prognostic nutritional index predicts survival and response to first-line chemotherapy in advanced biliary cancer. Liver Int. 2020 Mar;40(3):704-711; b) Inflammatory cells infiltrate and angiogenesis in locally advanced and metastatic cholangiocarcinoma. Eur J Clin Invest. 2019 May;49(5):e13087)

Author Response

We thank the editor and the reviewers for the thorough assessment of our manuscript. We have provided point-by-point responses to each of the comments beginning on the following page, and the revised portions of our manuscript are shown in colored text. We hope the changes made to our manuscript have made it more suitable for publication in your journal. We thank you for your consideration and look forward to hearing from you.

Reviewer 3

Comments and Suggestions for Authors

The manuscript entitled “Examination of Prognostic Factors Affecting Long term Survival of Patients with Stage 3/4 Gallbladder Cancer without Distant Metastasis” is an intrigue original retrospective study improving clinical practice in gallbladder cancer.

The Authors focus their attention on the role prognostic/predictive biomarkers for precision medicine in biliary cancers.

This paper is well written and balance, so far it should be published after some minor revisions: 

  • abbreviation should be checked

Thank you for your comment. We have checked the abbreviations throughout the manuscript.

  • please revised the mistyping

Thank you for your comment. We have had the manuscript proofread by an English-language professional editing service.

  • English should be revised

Thank you for your comment. We have had the manuscript proofread and reviewed by an English-language professional editing service again.

In univariate and multivariate analysis, Authors consider several factors. In any case, only few of them result statistically significant. Later, in Overall survival rate based on the number of prognostic factors predicted preoperative paragraph, the Authors write Further, the 5-year survival rate was 54.2% for 0 factors, 34.2% for 1 factor, and 4.42% for 2 or 3 94 factors, when liver invasion was 5 mm or more, left marginal invasion or the entire HDL, and 95 metastases of 4 or more regional lymph nodes were considered as predictable prognostic factors to 96 some extent before surgery (Figure 1 and Table S2A). The factors are not the same. Please specify.

Thank you for your comment.

As you pointed out, the univariate and multivariate analyses of the risk factors for overall survival showed that blood loss ≥1400 g (vs. <1400 g, hazard ratio [5]: 2.19), histologically poorly differentiated tumors or others (vs. well- or moderately-differentiated, HR: 2.26), liver invasion ≥ 5 mm (vs. no invasion, HR: 2.42), ≥4 regional lymph node metastases (vs. no lymph node metastasis [6], HR: 2.25), and surgery without adjuvant chemotherapy (vs. with, HR: 1.93) were independent risk factors. Invasion of the left margin or the entire area of the HDL (vs. no invasion, HR: 1.64, p=0.053) and postoperative morbidity with a Clavien-Dindo Classification ≥3 (vs. ≤2, HR: 1.64, p = 0.054) were marginally insignificant.

Among these factors, a number of tumor characteristics that can be predicted, to some extent, before surgery, including liver invasion ≥ 5 mm, invasion of the left margin or the entire area of the HDL, and ≥ 4 regional lymph node metastases, were used for analyzing the survival rate.

  • Figure 2 should be improved (it contains overlapping images)

Thank you for your comment.

Since this is a recurrence-free survival diagram, it is different from the overall survival diagram shown in Figure 1. Based on your comment, the figure showing recurrence-free survival has been moved to Supplementary Figure 2 because the figure shows a similar trend. Figure 2 now shows histological images to address the concerns of another reviewer.

  • Discussion: Authors should write a brief introduction on clinical/biological features of CCA. Please better the introduction with the informations required (they should refer to Current opinion on clinical practice diagnostic and therapeutic algorithms: A review of the literature and a long-standing experience of a referral center. Dig Liver Dis. 2016 Mar;48(3):231-41)

Thank you for your suggestion. Since the article referred to was about cholangiocarcinoma, I have, therefore, added another review article relevant for gallbladder cancer to the Discussion, which has been revised as follows:

L125: GBC is the most frequent biliary tract cancer according to autopsy studies; it is characterized by poor prognosis, with 5-year relative survival rates ranging from 2-70% depending on the stage of progression. GBC has a particularly high incidence in Chile, Japan, and northern India and is commonly associated with chronic inflammation, metaplasia progressing to dysplasia, carcinoma in situ, and then, invasive cancer. Gallstone disease, gallbladder polyps, chronic Salmonella infection, congenital biliary cysts, and abnormal pancreaticobiliary duct junctions have been reported as important precursors [7].

  • It might be interesting to understand the possibility of recovering data on the nutritional or inflammatory status of these patients, since they could represent prognostic factors for this pathology (Authors can refer to: a) The prognostic nutritional index predicts survival and response to first-line chemotherapy in advanced biliary cancer. Liver Int. 2020 Mar;40(3):704-711; b) Inflammatory cells infiltrate and angiogenesis in locally advanced and metastatic cholangiocarcinoma. Eur J Clin Invest. 2019 May;49(5):e13087)

Thank you for your valuable comment. We have added the following sentence to the Discussion section:

L198: Furthermore, it has been reported that nutritional status is associated with the effects and outcomes of chemotherapy [30], and that inflammatory cells in the tumor microenvironment support tumor growth [31]; therefore, maintenance of nutritional status and perioperative anti-inflammatory strategies may be important.